# ORDINARY DIFFERENTIAL EQUATIONS ON GRAPH NETWORKS

## ABSTRACT

Recently various neural networks have been proposed for irregularly structured data such as graphs and manifolds. To our knowledge, all existing graph networks have discrete depth. Inspired by neural ordinary differential equation (NODE) for data in the Euclidean domain, we extend the idea of continuous-depth models to graphs, and propose graph ordinary differential equation (GODE). The derivative of hidden node states are parameterized with a graph neural network, and the output states are the solution to this ordinary differential equation. Noticing that NODEs are typically trained with the adjoint method, with the advantages of adaptive evaluation and a free-form continuous invertible model; however, their performance on benchmark image classification tasks is significantly inferior to discrete-layer models. We show the reason is adjoint method generates inaccurate gradient due to numerical error in reverse-mode integration. We then propose a memory-efficient framework for free-form ODEs with accurate gradient estimation, which is fundamental to deep-learning models. Furthermore, when applied to invertible blocks, our method achieves constant memory cost with GODE (NODE). We show our method for free-form ODEs generalizes to various model structures and achieves high accuracy for both NODE and GODE in benchmark tasks.

## 1 INTRODUCTION

Convolutional neural networks (CNN) have achieved great success in various tasks, such as image classification (He et al., 2016) and segmentation (Long et al., 2015), video processing (Deng et al., 2014) and machine translation (Sutskever et al., 2014). However, CNNs are limited to data that can be represented by a grid in the Euclidean domain, such as images (2D grid) and text (1D grid), which hinders their application in irregularly structured datasets.

A graph data structure represents objects as nodes and relations between objects as edges. Graphs are widely used to model irregularly structured data, such as social networks (Kipf & Welling, 2016), protein interaction networks (Fout et al., 2017), citation and knowledge graphs (Hamaguchi et al., 2017). Early works use traditional methods such as random walk (Lovász et al., 1993), independent component analysis (ICA) (Hyvärinen & Oja, 2000) and graph embedding (Yan et al., 2006) to model graphs, however their performance is inferior due to the low expressive capacity.

Recently a new class of models called graph neural networks (GNN) (Scarselli et al., 2008) were proposed. Inspired by the success of CNNs, researchers generalize convolution operations to graphs to capture the local information. There are mainly two types of methods to perform convolution on a graph: spectral methods and non-spectral methods. Spectral methods typically first compute the graph Laplacian, then perform filtering in the spectral domain (Bruna et al., 2013). Other methods aim to approximate the filters without computing the graph Laplacian for faster speed (Defferrard et al., 2016). For non-spectral methods, the convolution operation is directly performed in the graph domain, aggregating information only from the neighbors of a node (Duvenaud et al., 2015; Atwood & Towsley, 2016). The recently proposed GraphSAGE (Hamilton et al., 2017) learns a convolution kernel in an inductive manner.

To our knowledge, all existing GNN models mentioned above have a structure of discrete layers. The discrete structure makes it hard for the GNN to model continuous diffusion processes (Freidlin & Wentzell, 1993; Kondor & Lafferty, 2002) in graphs. The recently proposed neural ordinary

differential equation (NODE) (Chen et al., 2018) views a neural network as an ordinary differential equation (ODE), whose derivative is parameterized by the network, and the output is the solution to this ODE. We extend NODE from the Euclidean domain to graphs and propose graph ordinary differential equations (GODE), where the message propagation on a graph is modeled as an ODE.

NODEs are typically trained with adjoint method. NODEs have the advantages of adaptive evaluation, accuracy-speed control by changing error tolerance, and are free-form continuous invertible models (Chen et al., 2018; Grathwohl et al., 2018). However, to our knowledge, in benchmark image classification tasks, NODEs are significantly inferior to state-of-the-art discrete-layer models (error rate: 19% for NODE vs 7% for ResNet18 on CIFAR10) (Dupont et al., 2019; Gholami et al., 2019). In this work, we show this is caused by error in gradient estimation during training of NODE, and propose a memory-efficient framework for accurate gradient estimation. We demonstrate our framework for free-form ODEs generalizes to various model structures, and achieves high accuracy for both NODE and GODE in benchmark tasks. Our contribution can be summarized as follows:

1. We propose a framework for free-form NODEs to accurately estimate the gradient, which is fundamental to deep-learning models. Our method significantly improves the performance on benchmark classification (reduces test error from 19% to 5% on CIFAR10).
2. Our framework is memory-efficient for free-form ODEs. When applied to restricted-form invertible blocks, the model achieves constant memory usage.
3. We generalize ODE to graph data and propose GODE models.
4. We demonstrate improved performance on different graph models and various datasets.

## 2 RELATED WORKS

### 2.1 NEURAL NETWORKS AND DIFFERENTIAL EQUATIONS

There have been efforts to view neural networks as differential equations. Lu (2017) viewed a residual network as a discretization of a differential equation and proposed several new architectures based on numerical methods in ODE solver. Haber & Ruthotto (2017) proposed a stable architecture based on analysis of the ODE. Chen et al. (2018) proposed neural ordinary differential equation (NODE), which treats the neural network as a continuous ODE. NODE was later used in a continuous normalizing flow for generative models (Grathwohl et al., 2018).

There have been many studies on the training of NODE. The adjoint method has long been widely used in optimal control (Stapor et al., 2018) and geophysical problems (Plessix, 2006), and recently applied to ODE (Chen et al., 2018). Dupont et al. (2019) proposed augmented neural ODEs to improve the expressive capacity of NODEs. However, to our knowledge, none of the methods above discusses the inaccurate gradient estimation issue; empirical performances of NODE in benchmark classification tasks are significantly inferior to state-of-the-art discrete-layer models.

### 2.2 GRAPH NEURAL NETWORKS

GNNs can be divided into two categories: spectral methods and non-spectral methods. Spectral GNNs perform filtering in the Fourier domain of a graph, thus need information of the whole graph to determine the graph Laplacian. In contrast, non-spectral GNNs only consider message aggregation around neighbor nodes, therefore are localized and generally require less computation (Zhou et al., 2018).

We first briefly introduce several spectral methods. Bruna et al. (2013) first introduced graph convolution in the Fourier domain based on the graph Laplacian, however the computation burden is heavy because of non-localized filters. Henaff et al. (2015) incorporated a graph estimation procedure in spectral networks and parameterized spectral filters into a localized version with smooth coefficients. Defferrard et al. (2016) used Chebyshev expansion to approximate the filters without the need to compute the graph Laplacian and its eigenvectors, therefore significantly accelerated the running speed. Kipf & Welling (2016) proposed to use a localized first-order approximation of graph convolution on graph data and achieved superior performance in semi-supervised tasks for node classification. Defferrard et al. (2016) proposed fast localized spectral filtering on graphs.

Non-spectral methods typically define convolution operations on a graph, only considering neighbors of a certain node. MoNet (Monti, 2017) uses a mixture of CNNs to generalize convolution to

graphs. GraphSAGE (Hamilton et al., 2017) samples a fixed size of neighbors for each node for fast localized inference. Graph attention networks (Veličković et al., 2017) learn different weights for different neighbors of a node. The graph isomorphism network (GIN) (Xu et al., 2018) has a structure as expressive as the Weisfeiler-Lehman graph isomorphism test.

## 2.3 Invertible Blocks

Invertible blocks are a family of neural network blocks whose forward function is a bijective mapping. Therefore, the input to a bijective block can be accurately reconstructed from its outputs. Invertible blocks have been used in normalizing flow (Rezende & Mohamed, 2015; Dinh, 2016; Kingma & Dhariwal, 2018; Dinh et al., 2014; Kingma et al., 2016), where the model is required to be invertible in order to calculate the log-density of data distribution. Later on, Jacobsen et al. (2018) used bijective blocks to build invertible networks. Gomez et al. (2017) proposed to use invertible blocks to perform back propagation without storing activation, which achieves a memory-efficient network structure. They were able to discard activation of middle layers, because each layer's activation can be reconstructed from the next layer with invertible blocks.

## 3 Accurate gradient estimation for training of NODE

### 3.1 From Discrete Models to Continuous Models

We first consider discrete-layer models with residual connection (He et al., 2016), which can be represented as:

$$x_{k+1} = x_k + f_k(x_k) \tag{1}$$

where $x_k$ is the states in the $k$th layer; $f_k(\cdot)$ is any differentiable function whose output has the same shape as its input.

When we add more layers with shared weights, and let the stepsize in Eq. 1 go to infinitesimal, the difference equation turns into a neural ordinary differential equation (NODE) (Chen et al., 2018):

$$\frac{\mathrm{d}z(t)}{\mathrm{d}t} = f(z(t), t) \tag{2}$$

We use $z(t)$ in the continuous case and $x_k$ in the discrete case to represent hidden states. $f(\cdot)$ is the derivative parameterized by a network. Note that a key difference between Eq. 1 and 2 is the form of $f$: in the discrete case, different layers (different $k$ values) have their own function $f_k$; while in the continuous case, $f$ is shared across all time $t$.

The forward pass of model with discrete layers can be written as:

$$x_0 = input, \quad x_1 = x_0 + f_0(x_0), \quad ..., \quad x_K = x_{K-1} + f_{K-1}(x_{K-1}) \tag{3}$$

where $K$ is the total number of layers. Then an output layer (e.g. fully-connected layer for classification) is applied on $x_K$.

The forward pass of a NODE is:

$$z(T) = z(0) + \int_{t=0}^{T} \frac{\mathrm{d}z(t)}{\mathrm{d}t} \mathrm{d}t = \text{input} + \int_{t=0}^{T} f(z(t), t) \mathrm{d}t \tag{4}$$

where $z(0) = $ input and $T$ is the integration time, corresponding to number of layers $K$ in the discrete case. The transformation of states $z$ is modeled as the solution to the NODE. Then an output layer is applied on $z(T)$. Integration in the forward pass can be performed with any ODE solver, such as the Euler Method, Runge-Kutta Method, VODE solver and Dopris Solver (Milne & Milne, 1953; Brown et al., 1989; Ascher et al., 1997).

### 3.2 Back-prop with Adjoint Method is Sensitive to Numerical Error

The adjoint method is widely used in optimal process control and functional analysis (Stapor et al., 2018; Pontryagin, 2018). We follow the method by (Chen et al., 2018). Denote model parameters as $\theta$, which is independent of time. Define the adjoint as:

$$a(t) = \frac{\partial L}{\partial z(t)} \tag{5}$$

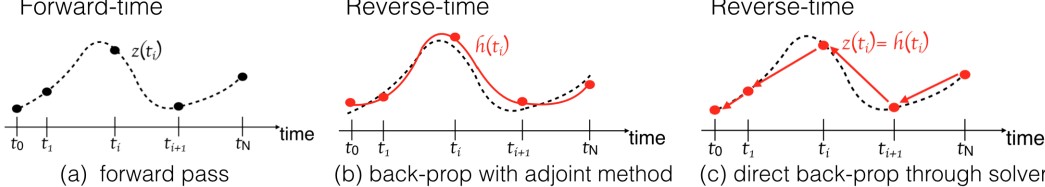

Figure 1: Comparison of two methods for back-propagation on NODE. As in figure (a), the ODE solver is discretized at points $\{t_0, t_1, ..., t_N\}$ during forward pass. Black dashed curve shows hidden state solved in forward-time, denoted as $z(t)$. Figure (b) shows the adjoint method, red solid line shows the hidden state solved in reverse-time, denoted as $h(t)$. Ideally $z(t) = h(t)$ and dashed curve overlaps with solid curve; however, the reverse-time solution could be numerically unstable, and causes $z(t) \neq h(t)$, thus causes error in gradient. Figure (c) shows the direct back-propagation through ODE solver. In direct back-propagation, we save evaluation time points $\{t_0, t_1, ...t_N\}$ during forward pass; during backward pass, we re-build the computation graph by directly evaluating at the same time points. In this way, $z(t_i) = h(t_i)$. Since the hidden state can be accurately reconstructed, the gradient can be accurately evaluated.

where $L$ is the loss function. Then we have

$$\frac{\mathrm{d}a(t)}{\mathrm{d}t} = -a(t)^T \frac{\partial f(z(t), t, \theta)}{z(t)}, \quad \frac{\mathrm{d}L}{\mathrm{d}\theta} = -\int_T^0 a(t)^T \frac{\partial f(z(t), t, \theta)}{\partial \theta} \mathrm{d}t \tag{6}$$

with detailed proof from optimization perspective in appendix F. Then we can perform gradient descent to optimize $\theta$ to minimize $L$. Eq. 6 is a reverse-time integration, which can be solved with any ODE solver (Chen et al., 2018). To evaluate $\frac{\partial f(z(t), t, \theta)}{\partial \theta}$, we need to determine $z(t)$ by solving Eq. 2 reverse-time (Directly storing $z(t)$ during forward pass requires a large memory consumption, because the continuous model is equivalent to an infinite-layer model). To summarize, in the forward pass we solve Eq. 2 forward in time; in the backward pass, we solve Eq. 2 and 6 reverse in time, with initial condition determined from Eq. 5 at time $T$.

We give an intuition why the reverse-time ODE solver causes inaccurate gradient in adjoint methods. The backward pass (Eq. 6) requires determining $f(z(t), t, \theta)$ and $\frac{\partial f(z(t), t, \theta)}{\partial \theta}$, which requires determining $z(t)$ by solving Eq. 2 reverse-time. As shown in Fig. 1 (a,b), the hidden state solved forward-time ($z(t_i)$) and the hidden state solved reverse-time ($h(t_i)$) may not be equal; this could be caused by the instability of reverse-time ODE, and is represented by the mismatch between $z(t)$ (dashed curve) and $h(t)$ (solid curve). Error $h(t) - z(t)$ will cause error in gradient $\frac{\mathrm{d}L}{\mathrm{d}\theta}$.

**Proposition 1** *For an ODE in the form $\frac{\mathrm{d}z(t)}{\mathrm{d}t} = f(z(t), t)$, denote the Jacobian of $f$ as $J_f$. If this ODE is stable both in forward-time and reverse-time, then $\mathrm{Re}(\lambda_i(J_f)) = 0 \; \forall i$, where $\lambda_i(J_f)$ is the ith eigenvalue of $J_f$, and $\mathrm{Re}(\lambda)$ is the real part of $\lambda$.*

Detailed proof is in appendix C. Proposition 1 indicates that if the Jacobian of the original system Eq. 2 has eigenvalues whose real-part are not 0, then either the reverse-time or forward-time ODE is unstable. When $|\mathrm{Re}(\lambda)|$ is large, either forward-time or reverse-time ODE is sensitive to numerical errors. This phenomenon is also addressed in Chang et al. (2018). This instability affects the accuracy of solution to Eq. 2 and 6, thus affects the accuracy of the computed gradient.

### 3.3 MEMORY-EFFICIENT DIRECT BACK-PROPAGATION THROUGH ODE SOLVER

The adjoint method might be sensitive to numerical errors when solving the ODE in reverse-time. To resolve this, we propose to directly back-propagate through the ODE solver.

As in Fig. 1(a), the ODE solver uses discretization for numerical integration, evaluated at time points $\{t_0, t_1, ...t_N\}$. Fig. 1(c) demonstrates the direct back-propagation with accurate hidden states $h(t_i)$, which can be achieved with two methods: (1) the activation $z(t_i)$ can be saved in cache for back-prop, but requires huge memory; or (2) we can accurately reconstruct $z(t_i)$ by re-building the computation graph directly at evaluated time points $\{t_i\}$. Since the model is evaluated at the same time points $t_i$ in forward-time, it's guaranteed that $z(t_i) = h(t_i)$. Therefore direct back-prop is accurate, regardless of the stability of Eq. 2.

Similar to the continuous case, we can define the adjoint with discrete time. Then we have:

$$a_i = \frac{\partial L}{\partial z(t_i)}, \quad a_{i+1} = a_i \frac{\partial z(t_{i+1})}{\partial z(t_i)}, \quad \frac{\mathrm{d}L}{\mathrm{d}\theta} = \sum_{i=1}^{N} a_i \frac{\partial z(t_i)}{\partial \theta} \tag{7}$$

where $a_i$ is the adjoint for the $i$th step in discrete forward-time ODE solution. Eq. 7 can be viewed as a numerical discretization of Eq. 6. We show Eq. 6 can be derived from an optimization perspective. Detailed derivations of Eq. 6-7 are in appendix E and F.

---

**Algorithm 1:** Algorithm for accurate gradient estimation in ODE solver for free-form functions

---

Define model $\frac{\mathrm{d}z(t)}{\mathrm{d}t} = f(z(t), t)$, where $f$ is a free-form function. Denote integration time as $T$.

**Forward** $(f, T, z_0, tolerance)$:
    $t = 0, z = z_0$
    $state_0 = f.state\_dict()$, cache.save($state_0$)
    Select initial step size $h = h_0$ (adaptively with adaptive step-size solver).
    $time\_points = empty\_list()$
    **While** $t < T$:
        $state = f.state\_dict(), \quad accept\_step = False$
        **While Not** $accept\_step$:
            $f.load\_state\_dict(state)$
            **with** $grad\_disabled$:
                $z\_new, error\_estimate = step(f, z, t, h)$
            **If** $error\_estimate < tolerance$:
                $accept\_step = True$
                $z = z\_new, \quad t = t + h, \quad time\_points.append(t)$
            **else**:
                reduce stepsize $h$ according to $error\_estimate$
                **delete** $z\_new, error\_estimate$ and related local computation graph
    cache.save($time\_points$)
    **return** $z$, cache

**Backward** $(f, T, z_0, tolerance, cache)$:
    $\{t_0, t_1, t_2, ...t_{N-1}, t_N\} = cache.time\_points$
    **For** $t_i$ in $\{t_0, t_1, t_2, ...t_{N-1}, t_N\}$ :
        $z(t_{i+1}) = step(f, z(t_i), t_i, step = t_{i+1} - t_i)$
    **For** $t_i$ in $\{t_N, t_{N-1}, ..., t_1, t_0\}$ :
        Determine $a(t_i) = \frac{\partial L}{\partial z(t_i)}$ and $\frac{\partial z(t_i)}{\partial \theta}$
    $\frac{\mathrm{d}L}{\mathrm{d}\theta} = \sum_{i=1}^{N} a_i \frac{\partial z(t_i)}{\partial \theta}$
    **return** $\frac{\mathrm{d}L}{\mathrm{d}\theta}$

---

Details of our method are summarized in Algorithm 1. We discuss its properties below:

**Summary of the algorithm** (1) During forward pass, the solver performs a numerical integration, with the stepsize adaptively varying with error estimation. (2) During forward pass, the solver outputs the integrated value, and the evaluation time points $\{t_i\}$. All middle activations are deleted to save memory. (3) During backward pass, the solver re-builds the computation graph, by **directly** evaluating at saved time points, without adaptive searching. (4) During backward pass, the solver performs a numerical version (Eq. 7) of reverse-time integration (Eq. 6).

**Support for free-form continuous dynamics** There's no constraint on the form of $f$. Therefore, our algorithm is a generic method.

**Memory consumption analysis** (1) Suppose $f$ has $N_f$ layers, the number of forward evaluation step is $N_t$ on average, and the evaluations to adaptively search for an optimal stepsize is $K$. A naive solver will take $O(N_f \times N_t \times K)$, while our method consumes $O(N_f \times N_t)$ because all middle

activations are deleted during forward pass, and we don't need to search for optimal stepsize in backward pass. (2) If we perform step-wise checkpoint method, where we only store $z(t_i)$ for all $t_i$, and compute the gradient $\frac{\partial z(t_{i+1})}{\partial z(t_i)}$ for one $t_i$ at a time, then the memory consumption can be reduced to $O(N_f+N_t)$. (3) Since the solver can handle free-form functions, it can also handle restricted form invertible block (see below). In this case, we don't need to store $z(t_i)$, and the memory consumption can reduce to $O(N_f)$.

**More memory-efficient solver with invertible blocks**  Restricting the form of $f$ to invertible blocks (Gomez et al., 2017) allows for $O(N_f)$ memory consumption. For invertible blocks, input $x$ is split into two parts $(x_1, x_2)$ of the same size (e.g. $x$ has shape $N \times C$, where $N$ is batch size, $C$ is channel number; we can split $x$ into $x_1$ and $x_2$ with shape $N \times \frac{C}{2}$). The forward and inverse of a bijective block can be denoted as:

$$\begin{cases} y_2 = \psi\Big(x_2, F(x_1)\Big) \\ y_1 = \psi\Big(x_1, G(y_2)\Big) \end{cases} \begin{cases} x_1 = \psi^{-1}\Big(y_1, G(y_2)\Big) \\ x_2 = \psi^{-1}\Big(y_2, F(x_1)\Big) \end{cases} \tag{8}$$

where the output of a bijective block is denoted as $(y_1, y_2)$ with the same size as $(x_1, x_2)$. $F$ and $G$ are any differentiable neural networks, whose output has the same shape as the input. $\psi(\alpha, \beta)$ is a differentiable bijective function $w.r.t$ $\alpha$ when $\beta$ is given; $\psi^{-1}(\alpha, \beta)$ is the inverse function of $\psi$.

**Theorem 1** *If $\psi(\alpha, \beta)$ is a bijective function $w.r.t$ $\alpha$ when $\beta$ is given, then the block defined by Eq. 8 is a bijective mapping.*

Proof of Theorem 1 is given in appendix D. Based on this, we can apply different $\psi$ functions for different tasks. Since $x$ can be accurately reconstructed from $y$, there's no need to store activations, hence is memory-efficient. Details for back-prop without storing activation are in appendix B.

## 4    NEURAL ORDINARY DIFFERENTIAL EQUATION ON GRAPH NETWORKS

We first introduce graph neural networks with discrete layers, then extend to the continuous case and introduce graph ordinary differential equations (GODE).

### 4.1    MESSAGE PASSING IN GNN

As shown in Fig. 2, a graph is represented with nodes (marked with circles) and edges (solid lines). We assign a unique color to each node for ease of visualization. Current GNNs can generally be represented in a message passing scheme (Fey & Lenssen, 2019):

$$message_{(v,u)} = \phi^{(k)}(x_{k-1}^u, x_{k-1}^v, \mathbf{e}_{u,v}) \tag{9}$$

$$aggregation_u = \zeta_{v \in \mathcal{N}(u)}(message_{(v,u)}) \tag{10}$$

$$x_k^u = \gamma^{(k)}(x_{k-1}^u, aggregation_u) \tag{11}$$

where $x_k^u$ represents states of the $u$th node in the graph at $k$th layer and $\mathbf{e}_{u,v}$ represents the edge between nodes $u$ and $v$. $\mathcal{N}(u)$ represents the set of neighbor nodes for node $u$. $\zeta$ represents a differentiable, permutation invariant operation such as $mean$, $max$ or $sum$. $\gamma^{(k)}$ and $\phi^{(k)}$ are differentiable functions parameterized by neural networks.

For a specific node $u$, a GNN can be viewed as a 3-stage model, corresponding to Eq. 9-11: (1) Message passing, where neighbor nodes $v \in \mathcal{N}(u)$ send information to node $u$, denoted by $message_{(v,u)}$. The message is generated from function $\phi(\cdot)$, parameterized by a neural network. (2) Message aggregation, where a node $u$ aggregates all messages from its neighbors $\mathcal{N}(u)$, denoted as $aggregation_u$. The aggregation function $\zeta$ is typically permutation invariant operations such as $mean$ and $sum$, because graphs are invariant to permutation. (3) Update, where the states of a node are updated according to its original states $x_{k-1}^u$ and aggregation of messages $aggregation_u$, denoted as $\gamma(\cdot)$.

### 4.2 CONTINUOUS-TIME MODELS ON GRAPHS

We can convert a discrete-time GNN to continuous-time GNN by replacing $f$ in Eq. 2 with the message passing process defined in Eq. 9 to 11, which we call graph ordinary differential equation (GODE). A diagram of GODE is shown in Fig. 2. Because GODE is an ODE in nature, it can capture highly non-linear functions, thus has the potential to outperform its discrete-layer counterparts.

We demonstrate that the asymptotic stability of GODE could be related to the over-smoothing phenomena (Li et al., 2018). It's demonstrated that graph convolution is a special case of Laplacian smoothing (Li et al., 2018), which can be written as $Y = (I - \gamma \tilde{D}^{-1/2} \tilde{L} \tilde{D}^{-1/2})X$ where $X$ and $Y$ are the input and output of a graph-conv layer respectively, $\tilde{A} = A + I$ where $A$ is the adjacency matrix, and $\tilde{D}$ is the corresponding degree matrix of $\tilde{A}$, and $\gamma$ is a positive scaling constant.

When modified from a discrete model to a continuous model, the continuous smoothing process is:

$$\frac{dX}{dt} = -\gamma \tilde{D}^{-1/2} \tilde{L} \tilde{D}^{-1/2} X \tag{12}$$

Since all eigenvalues of the symmetrically normalized Laplacian are real and non-negative, then all eigenvalues of the above ODE are real and non-positive. Suppose all eigenvalues of the normalized Laplacian are non-zero. In this case, the ODE has only negative eigenvalues, hence the ODE above is $asymptotically\ stable$ (Lyapunov, 1992). Hence as time $t$ grows sufficiently large, all trajectories are close enough. In the experiments, this suggests if integration time $T$ is large enough, all nodes (from different classes) will have very similar features, thus the classification accuracy will drop.

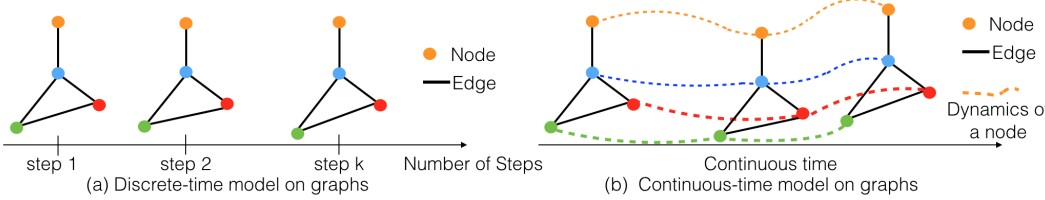

Figure 2: Diecrete-time and continuous-time models on a graph. Nodes are represented with circles, and each node is represented with a unique color. Edges are represented with solid lines. For discrete-time models in (a), the hidden states of nodes are updated with discrete steps. For continuous-time models in (b), hidden states of each node evolves continuously with time. The dynamics of nodes are represented with dashed lines, with the same color as corresponding nodes.

## 5 EXPERIMENTS

### 5.1 DATASETS

To evaluate our method on general NODE, we conducted experiments with a CNN-NODE on two benchmark image classification tasks (CIFAR10 and CIFAR100) (Krizhevsky et al., 2009).

We also evaluated our method on benchmark graph datasets, including 2 bioinformatic graph classification datasets (MUTAG and PROTEINS), 2 social network graph classification datasets (IMDB-BINRAY, REDDIT-BINARY) (Yanardag & Vishwanathan, 2015), and 3 citation networks (Cora, CiteSeer and PubMed). For graph classification tasks, different from the experiment settings in Xu et al. (2018), we input the raw dataset into our models without pre-processing. For node classification tasks, we performed transductive inference and strictly followed the train-validation-test split by Kipf & Welling (2016), where less than 6% nodes are used as training examples. Details of datasets are summarized in appendix A.

### 5.2 MODEL STRUCTURES

For image classification tasks, we directly modify a ResNet18 into its corresponding NODE model. For each block, the function is $\frac{dz(t)}{dt} = f(z(t))$ where $f$ is a sequence of $conv - bn - relu - conv - bn - relu$ layers. $f$ is the same as residual branch in ResNet, and it can be replaced with any free-form functions.

| | Ours | | | | | | Literature | | | |
|---|---|---|---|---|---|---|---|---|---|---|
| | Adaptive Stepsize Solvers | | | Fixed stepsize solvers | | | adjoint-ODE | ResNet18 | ResNet50 | ResNet101 |
| | Heun-Euler | RK23 | RK45 | Euler | RK2 | RK4 | | | | |
| CIFAR10 | 4.85 | 4.92 | 5.29 | 5.52 | 5.27 | 5.24 | 19.2 | 6.98 | 6.38 | 6.25 |
| CIFAR100 | 22.66 | 24.13 | 23.56 | 24.44 | 24.44 | 24.43 | 37.6 | 27.08 | 25.73 | 24.84 |

Table 1: **Error rate** on test set. "Ours" represent NODE models directly modified from ResNet18, trained with Heun-Euler solver, but tested with different solvers. "Adjoint" is the result when trained and tested with adjoint method, reported by (Gholami et al., 2019). We also report results from standard ResNet.

| | MUTAG | PROTEINS | IMDB | REDDIT |
|---|---|---|---|---|
| adjoint | 68.1± 4.6 | 67.0±3.7 | 72.1±0.4 | 69.5±5.9 |
| ours | **80.8±8.3** | **73.9±3.1** | **74.6±5.1** | **92.4±2.1** |

Table 2: **Accuracy** of adjoint method and direct back-prop, for a GODE model with GCN as the derivative function.

For tasks on graph datasets, GODE can be applied to any graph neural network by simply replacing $f$ in Eq. 2 with corresponding structures (free-form functions), or replacing $F, G$ in Eq. 8 with other structures (invertible blocks). To demonstrate that GODE is easily generalized to existing structures, we used several different GNN architectures, including the graph convolutional network (GCN) (Kipf & Welling, 2016), graph attention network (GAT) (Veličković et al., 2017), graph network approximated with Chebyshev expansion (ChebNet) (Defferrard et al., 2016), and graph isomorphism network (GIN) (Xu et al., 2018). For a fair comparison, we trained GNNs with different depths of layers (1-3 middle layers, besides an initial layer to transform data into specified channels, and a final layer to generate prediction), and reported the best results among all depths for each model structure.

On the same task, different models use the same hyper-parameters on model structures, such as channel number. For graph classification tasks, we set the channel number of hidden layers as 32 for all models; for ChebNet, we set the number of hops as 16. For node classification tasks, we set the channel number as 16 for GCN and ChebNet, and set number of hops as 3 for ChebNet; for GAT, we used 8 heads, and set each head as 8 channels.

For every GNN structure, we experimented with different number of hidden layers (1,2,3), and calculated the mean and variance of accuracy of 10 runs.

## 5.3 COMPARISON OF BACK-PROPAGATION METHODS

We compared the adjoint method and direct back-propagation on the same network, and demonstrated direct back-prop generates higher accuracy. For CNN-NODE on classification tasks, we directly modify a ResNet18 into NODE18, and report resuls in Table. 1; for graph networks, we train a GODE model with a GCN to parameterize the derivative, and report results in Table. 2.

**Empirical performance**  Direct back-propagation consistently outperformed the adjoint method for both tasks. This result validates our analysis on the instability of the adjoint method, which is intuitively caused by the instability of the reverse-time ODE. On image classification tasks, compared to adjoint method, our training method reduces error rate of NODE18 from 19% (37%) to 5%(23%) on CIFAR10 (CIFAR100). Furthermore, NODE18 has the same number of parameters as ResNet18, but outperforms deeper networks such as ResNet101 on both datasets. Our method also consistently outperforms the adjoint method on several benchmark graph datasets, as shown in Table. 2.

**Robustness to ODE solvers**  We implemented adaptive ODE solvers of different orders, as shown in Table 1. HeunEuler, RK23, RK45 are of order 1, 2, 4 respectively, i.e., for each step forward in time $f$ is evaluated 1, 2, 4 times respectively. During inference, using different solvers is equivalent to changing model depth (**without** re-training the network): for discrete-layer models, it generally causes huge error; for continuous models, we observe only around 1% increase in error rate. This suggests our method is robust to different orders of ODE solvers.

**Support for free-form functions**  Our method supports NODE and GODE models with free-form functions; for example, $f$ in NODE18 in Table. 1 is a free-form function.

| Model | | $\psi$ | Cora | CiteSeer | PubMed |
|---|---|---|---|---|---|
| GCN | DISC | | 81.6±0.5 | 71.6±0.3 | 79.2±0.1 |
| | ODE | add | 81.7±0.7 | 72.4±0.6** | 80.0±0.2** |
| | | l_sig | 81.8±0.3** | 72.4±0.8** | 80.1±0.3** |
| GAT | DISC | | 82.9±0.3 | 71.7±0.8 | 78.9±0.3 |
| | ODE | add | 83.3±0.3** | 72.1±0.6** | 79.1±0.5* |
| | | l_sig | 83.1±0.4* | 72.1±0.3** | 79.0±0.5 |
| Cheb | DISC | | 82.1±0.5 | 70.8±0.5 | 76.6±0.8 |
| | ODE | add | 82.4±0.5* | 71.1±0.5** | 77.8±1.2** |
| | | l_sig | 82.2±0.4* | 70.8±0.6 | 77.0±1.1* |

| Model | | | MUTAG | PROTEINS | IMDB | REDDIT |
|---|---|---|---|---|---|---|
| GCN | DISC | | 73.3±5.2 | 72.4±3.1 | 74.2±3.6 | 85.9±1.8 |
| | ODE | INV | 78.1±6.2** | 74.7±4.3** | 75.3±5.3* | 89.2±3.2** |
| | | free | 75.1±5.3** | 76.6±3.9** | 73.9±4.6 | 88.5±3.0** |
| Cheb | DISC | | 84.0±6.4 | 70.6±3.9 | 71.9±3.8 | 91.0±1.5 |
| | ODE | INV | 85.0±8.3 | 72.7±3.7** | 72.0±5.0* | 91.2±1.5 |
| | | free | 86.1±6.3* | 72.5±4.7** | 73.6±4.0** | 92.4±1.6** |
| GIN | DISC | | 85.0±6.4 | 73.0±3.1 | 73.3±5.1 | 89.2±2.5 |
| | ODE | INV | 87.7±5.2* | 74.5±3.5* | 75.4±3.9** | 90.5±1.5** |
| | | free | 87.8±5.1** | 73.1±3.7 | 74.8±4.1** | 91.0±2.4** |

Table 3: Results on node classification tasks. We compared various discrete-layer structures (marked with DISC) and their corresponding GODE models (marked with ODE). We tested GODE model with different $\psi$ functions ("l_sig" represents linear_sigmoid). For each model, we use * (**) to mark GODE models that outperform corresponding discrete-layer baselines at a 5% (1%) significance level under paired t-test.

Table 4: Results on graph-classification tasks. For each base model structure, discerete-layer model is marked with DISC; for corresponding GODE, we tested both free-form functions ("free"), and their invertible block form ("INV"). For each model, we use * (**) to mark GODE models that outperform corresponding discrete-layer baselines at a 5% (1%) significance level under paired t-test.

| Integration time | 0.5 | 1.0 | 1.5 | 2.0 | 5.0 | 10.0 | 20.0 | 100.0 |
|---|---|---|---|---|---|---|---|---|
| Cora | 80.5 | 81.6 | 80.1 | 80.1 | 79.3 | 77.6 | 68.9 | 35.1 |
| CIFAR10 | 91.3 | 95.2 | 94.2 | 88.4 | 10.0 | 10.0 | - | - |

Table 5: Accuracy of a free-form GCN-ODE on Cora and NODE18 on CIFAR10, varying with integration time.

## 5.4 GENERAL BIJECTIVE BLOCKS

We demonstrate that bijective blocks defined as Eq. 8 can be easily generalized: $F$ and $G$ are general neural networks, which can be adapted to different tasks; $\psi(\alpha, \beta)$ can be any differentiable bijective mapping $w.r.t.$ $\alpha$ when $\beta$ is given. We demonstrate two examples of $\psi$: (1) additive, forward is $\eta = \psi(\alpha, \beta) = \alpha + \beta$, inverse is $\alpha = \psi^{-1}(\eta, \beta) = \eta - \beta$; (2) linear_sigmoid, forward is $\eta = \psi(\alpha, \beta) = \alpha \times \text{sigmoid}(\beta)$, inverse is $\alpha = \psi^{-1}(\eta, \beta) = \eta/\text{sigmoid}(\beta)$.

Results for different $\psi$ are reported in Table 3. Note that we experimented with different depths and reported the best accuracy for each model, and performed a paired t-test on results from GODE and their discrete-layer counterparts. Most GODE models outperformed their corresponding discrete-layer models significantly, validating the effectiveness of GODE; different $\psi$ functions behaved similarly on our node classification tasks, indicating the continuous-time model is more important than coupling function $\psi$. We also validate the lower memory cost, with details in appendix B.

## 5.5 RESULTS ON GRAPH CLASSIFICATION TASK

Results for different models on graph classification tasks are summarized in Table 4. We experimented with different structures, including GCN, ChebNet and GIN; for corresponding GODE models (marked with ODE), we tested both free-form (marked with "free") and invertible block (marked with "INV"). We performed paired t-test comparing GODE and its discrete-layer counterparts. For most experiments, GODE models performed significantly better. This indicates the continuous process model might be important for graph models.

## 5.6 IMPACT OF INTEGRATION TIME

For a NODE and GODE model, during inference, we test the influence of integration time. Results are summarized in Table. 5. When integration time is short, the network does not gather sufficient information from neighbors; when integration time is too long, the model is sensitive to over-smooth issue, as discussed in Sec. 4.2. We observe accuracy drop in both cases.

## 6 CONCLUSIONS

We propose GODE, which enables us to model continuous diffusion process on graphs. We propose a memory-efficient direct back-propagation method to accurately determine the gradient for general free-form NODEs, and validate its superior performance on both image classification tasks and graph data. Furthermore, we related the over-smoothing of GNN to asymptotic stability of ODE. Our paper tackles the fundamental problem of gradient estimation for NODE; to our knowledge, it's the first paper to improve accuracy on benchmark tasks to comparable with state-of-the-art discrete layer models. It's an important step to apply NODE from theory to practice.

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

## A  DATASETS

We perform experiments on various datasets, including citation networks (Cora, CiteSeer, PubMed), social networks (COLLAB, IMDB-BINARY, REDDIT-BINARY), and bioinformatics datasets (MUTAG, PROTEINS). Details of each dataset are summarized in Table 1.

Table 1: Statistics of datasets

| Dataset | Graphs | Nodes | Edges | Features | Classes | Label rate |
|---|---|---|---|---|---|---|
| Cora | 1 | 2,708 | 5,278 | 1,433 | 7 | 0.052 |
| CiteSeer | 1 | 3,327 | 4,552 | 3,703 | 6 | 0.036 |
| PubMEd | 1 | 19,717 | 44,324 | 500 | 3 | 0.003 |
| MUTAG | 188 | 17.93 | 19.79 | 7 | 2 | 0.8 |
| PROTEINS | 1,113 | 39.06 | 72.82 | 3 | 2 | 0.8 |
| IMDB-BINARY | 1,000 | 19.77 | 96.53 | - | 2 | 0.8 |
| REDDIT-BINARY | 200 | 429.63 | 497.76 | - | 2 | 0.8 |

## B  DETAILS ABOUT INVERTIBLE BLOCKS

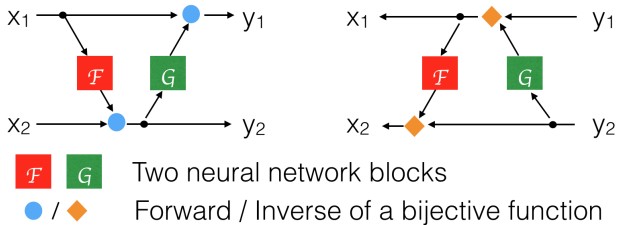

Figure 1: Structure of bijective blocks. $F$ and $G$ can be any differentiable neural network whose output has the same shape as its input. Blue dot (Orange diamond) represents the forward (inverse) of a bijective function, corresponding to $\psi$ ($\psi^{-1}$) in Eq. 8 of the main paper. Left (right) figure represents the forward (inverse) as in Eq. 8.

We explain the structure and conduct experiments for the invertible block here.

**Structure of invertible blocks**  Structure of invertible blocks are shown in Fig. 1. We follow the work of Gomez et al. (2017) with two important modifications: (1) We generalize to a family of bijective blocks with different $\psi$ in Eq. 8 in the main paper, while Gomez et al. (2017) restrict the form of $\psi$ to be $sum$. (2) We propose a parameter state checkpoint method, which enables bijective blocks to be called more than once, while still generating accurate inversion.

The algorithm is summarized in Algo. 2. We write the pseudo code for forward and backward function as in PyTorch. Note that we use "inversion" to represent reconstructing input from the output, and use "backward" to denote calculation of the gradient. To reduce memory consumption, in the forward function, we only keep the outputs $y_1, y_2$ and delete all other variables and computation graphs. In the backward function, we first "inverse" the block to calculate $x_1, x_2$ from $y_1, y_2$, then perform a local forward and calculate the gradient $\frac{\partial[y_1, y_2]}{\partial[x_1, x_2]}$.

**Experiments**  In this section we demonstrate that our bijective block is memory efficient. We trained a GODE model with bijective blocks, and compared the memory consumption using our memory-efficient function as in Algo. 2 and a memory-inefficient method as in conventional back-propagation. Results were measured with a batchsize of 100 on MUTAG dataset.

| Depth | Memory-efficient | Conventional |
|---|---|---|
| 10 | 2.2G | 5.3G |
| 20 | 2.6G | 10.5G |

Table 2: Memory consumption of bijective blocks. "Conventional" represents storing activation of all layers in cache, "memory-efficient" represents our method in Algo. 2.

Results are summarized in Table. 2. We measured the memory consumption with different depths, which is the number of ODE blocks. When depth increases from 10 to 20, the memory by conventional methods increases from 5.3G to 10.5G, while our memory-efficient version only increases from 2.2G to 2.6G. In theory, our bijective block takes $\mathcal{O}(1)$ memory, because we only need to store the outputs in cache, while deleting activations of middle layers. For memory-efficient network, the slightly increased memory consumption is because states of $F, G$ need to be cached; but this step takes up minimal memory compared to input data.

---

**Algorithm 2:** Function for memory-efficient bijective blocks

---

**Forward** (cache, $x_1, x_2, F, G, \psi$)
    **cache.save**([$F$ states, $G$ states])
    **forward in Eq. 8**
        $\eta_1 = F(x_1), \; y_2 = \psi(x_2, \eta_1)$
        $\eta_2 = G(y_2), \; y_1 = \psi(x_1, \eta_2)$
    **delete** $\eta_1, \eta_2, x_1, x_2$
    **delete** computation graphs generated by $F$ and $G$
    **return** cache, $y_1, y_2$

**Backward**(cache, $y_1, y_2, F, G, \psi, \frac{\partial L}{\partial y_1}, \frac{\partial L}{\partial y_2}$)
    **Reset** $F$ and $G$ states from cache
    **Inverse from** $y_1, y_2$ **to** $x_1, x_2$
        $\eta_2 = G(y_2), \; x_1 = \psi^{-1}(y_1, \eta_2)$
        $\eta_1 = F(x_1), \; x_2 = \psi^{-1}(y_2, \eta_1)$
    **Local forward pass and gradient**
        $X_1, X_2 = x_1.detach(), x_2.detach()$
        calculate $Y_1, Y_2$ from $X_1, X_2$ as Eq. 8
        determine $\partial[Y_1, Y_2]/\partial[X_1, X_2, \theta_F, \theta_G]$
        $\frac{\partial L}{\partial[x_1, x_2]} = \frac{\partial L}{\partial[y_1, y_2]} \frac{\partial[Y_1, Y_2]}{\partial[X_1, X_2]}$
        $\frac{\partial L}{\partial[\theta_F, \theta_G]} = \frac{\partial L}{\partial[y_1, y_2]} \frac{\partial[Y_1, Y_2]}{\partial[\theta_F, \theta_G]}$
    **delete** $Y_1, Y_2, X_1, X_2$
    **return** $\partial L/\partial[x_1, x_2], \partial L/\partial[\theta_F, \theta_G]$

---

## C  PROOF FOR PROPOSITION 1

**Proposition 1** *For an ODE in the form $\frac{\mathrm{d}z(t)}{\mathrm{d}t} = f(z(t), t)$, denote the Jacobian of $f$ as $J_f$. If this ODE is stable both in forward-time and reverse-time, then $\mathrm{Re}(\lambda_i(J_f)) = 0 \;\; \forall i$, where $\lambda_i(J_f)$ is the ith eigenvalue of $J_f$, and $\mathrm{Re}(\lambda)$ is the real part of $\lambda$.*

**Proof**  Denote $s = T - t$, where $T$ is the end time. Notice that the reverse-time in $t$ is equivalent to forward-time in $s$.
Therefore, we have forward-time ODE:

$$\frac{\mathrm{d}z(t)}{\mathrm{d}t} = f(z(t), t) \tag{1}$$

and reverse-time ODE:

$$\frac{\mathrm{d}z(s)}{\mathrm{d}s} = -f(z(s), s) = g(z(s), s) \tag{2}$$

Therefore, we have $\lambda(J_f) = -\lambda(J_g)$. For both forward-time and reverse-time ODE to be stable, the eigenvalues of $J$ need to have non-positive real part.
Therefore

$$\mathrm{Re}\lambda_i(J_f) \leq 0, \;\; \mathrm{Re}\lambda_i(J_g) = -\mathrm{Re}\lambda_i(J_f) \leq 0, \;\; \forall i \tag{3}$$

The only solution is

$$\mathrm{Re}\lambda_i(J_g) = -\mathrm{Re}\lambda_i(J_f) = 0, \;\; \forall i \tag{4}$$

## D  PROOF FOR THEOREM 1

**Theorem 1** *For bijective block whose forward and reverse mappings are defined as*

$$Forward(x_1, x_2) = \begin{cases} y_2 = \psi\Big(x_2, F(x_1)\Big) \\ y_1 = \psi\Big(x_1, G(y_2)\Big) \end{cases} \quad Reverse(y_1, y_2) = \begin{cases} x_1 = \psi^{-1}\Big(y_1, G(y_2)\Big) \\ x_2 = \psi^{-1}\Big(y_2, F(x_1)\Big) \end{cases}$$

*If $\psi(\alpha, \beta)$ is a bijective function w.r.t $\alpha$ when $\beta$ is given, then the block is a bijective mapping.*

**Proof**    To prove the forward mapping is bijective, it is equivalent to prove the mapping is both injective and surjective.

**Injective**    We need to prove, if $Forward(x_1, x_2) = Forward(x_3, x_4)$, then $x_1 = x_3, \ x_2 = x_4$.

The assumption above is equivalent to

$$Forward(x_1, x_2) = Forward(x_3, x_4) \iff y_2 = \psi(x_2, F(x_1)) = \psi(x_4, F(x_3)) \qquad (5)$$
$$\psi(x_1, G(y_2)) = \psi(x_3, G(y_2)) \qquad (6)$$

Since $\psi(\alpha, \beta)$ is bijective $w.r.t \ \alpha$ when $\beta$ is given, from Eq.(6), we have $x_1 = x_3$.
Similarly, condition on $x_1 = x_3$ and Eq.(5), using bijective property of $\psi$, we have $x_2 = x_4$.
Therefore, the mapping is injective.

**Surjective**    We need to prove $\forall \ [y_1, y_2], \exists \ [x_1, x_2] \ s.t. \ Forward(x_1, x_2) = [y_1, y_2]$.

Given $y_1, y_2$, we construct

$$x_1 = \psi^{-1}\Big(y_1, G(y_2)\Big), x_2 = \psi^{-1}\Big(y_2, F(x_1)\Big) \qquad (7)$$

Then for the forward function, given bijective property of $\psi$, apply $Forward$ and $Reverse$ defined in the proposition statement,

$$z_2 = \psi(x_2, F(x_1)) = \psi\Big(\psi^{-1}\big(y_2, F(x_1)\big), F(x_1)\Big) = y_2 \qquad (8)$$

$$z_1 = \psi(x_1, G(y_2)) = \psi\Big(\psi^{-1}\big(y_1, G(y_2)\big), G(y_2)\Big) = y_1 \qquad (9)$$

Therefore we construct $x_1, x_2 \ s.t. \ Forward(x_1, x_2) = [y_1, y_2]$.
Therefore the mapping is surjective.
Therefore is bijective.

## E    DERIVATION OF GRADIENT IN DISCRETE CASE

We use a figure to demonstrate the computation graph, and derive the gradient from the computation graph.

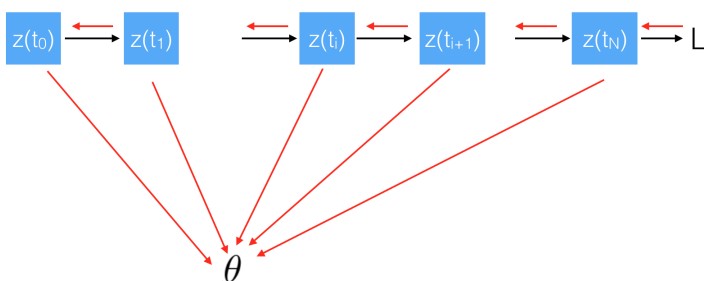

The loss is $L$, forward pass is denoted with black arrows, gradient back-propagation is shown with red arrows. We use $p$ to denote each path from $\theta$ to $L$, corresponding to all paths in red that goes from $L$ to $\theta$.

$$\frac{\mathrm{d}L}{\mathrm{d}\theta} = \sum_p \frac{\partial L_p}{\partial \theta} = \sum_i \frac{\partial L}{\partial z(t_i)} \frac{\partial z(t_i)}{\partial \theta} = \sum_i a_i \frac{\partial z(t_i)}{\partial \theta} \qquad (10)$$

$$a_i = \frac{\partial L}{\partial z(t_i)} = \frac{\partial L}{\partial z(t_{i+1})} \frac{\partial z(t_{i+1})}{\partial z(t_i)} = a_{i+1} \frac{\partial z(t_{i+1})}{\partial z(t_i)} \tag{11}$$

## F    DERIVATION OF PARAMETER GRADIENTS IN CONTINUOUS CASE

In this section we derive the gradient of parameters in an neural-ODE model from an optimization perspective. Then we extend from continuous cases to discrete cases.

**Notations**    With the same notations as in the main paper, we use $z(t)$ to denote hidden states $z$ at time $t$. Denote parameters as $\theta$, and input as $x$, target as $y$, and predicted output as $\hat{y}$. Denote the loss as $J(\hat{y}, y)$. Denote the integration time as 0 to $T$.

**Problem setup**    The continuous model is defined to follow an ODE:

$$\frac{dz(t)}{dt} = f(z(t), t, \theta), \ \ s.t. \ z(0) = x \tag{12}$$

We assume $f$ is differentiable, since $f$ is represented by a neural network in our case. The forward pass is defined as:

$$\hat{y} = z(T) = z(0) + \int_0^T f(z(t), t, \theta) dt \tag{13}$$

The loss function is defined as:

$$J(\hat{y}, y) = J(z(T), y) \tag{14}$$

We formulate the training process as an optimization problem:

$$\text{argmin}_\theta \frac{1}{N} \sum_{i=1}^N J(\hat{y}_i, y_i) \ \ s.t. \ \frac{dz(t)}{dt} = f(z(t), t, \theta), \ \ z_i(0) = x_i \tag{15}$$

For simplicity, Eq. 15 only considers one ODE block. In the case of multiple blocks, $z(T)$ is the input to the next ODE block. As long as we can derive $\frac{dLoss}{d\theta}$ and $\frac{dLoss}{dz(0)}$ when $\frac{dLoss}{dz(T)}$ is given, the same analysis here can be applied to the case with a chain of ODE blocks.

**Lagrangian Multiplier Method**    We use the Lagrangian Multiplier Method to solve the problem defined in Eq. 15. For simplicity, only consider one example (can be easily extended to multiple examples cases), the Lagrangian is

$$L = J(z(T), y) + \int_0^T \lambda(t)[\frac{dz(t)}{dt} - f(z(t), t, \theta)] dt \tag{16}$$

Karush-Kuhn-Tucker (KKT) conditions are necessary conditions for an solution to be optimal. In the following sections we start from the KKT condition and derive our results.

**Derivative w.r.t. $\lambda$**    At optimal point, we have $\frac{\delta L}{\delta \lambda} = 0$. Note that $\lambda$ is a function of $t$, we derive the derivative from calculus of variation.

Consider a cotninuous and differentiable perturbation $\overline{\lambda(t)}$ on $\lambda(t)$, and a scalar $\epsilon$, $L$ now becomes a function of $\epsilon$,

$$L(\epsilon) = J(z(0) + \int_0^T f(z(t), t, \theta), y) + \int_0^T (\lambda(t) + \epsilon \overline{\lambda(t)})[\frac{dz(t)}{dt} - f(z(t), t, \theta)] dt \tag{17}$$

It's easy to check the conditions for Leibniz integral rule, and we can switch integral and differentiation, thus:

$$\frac{dL}{d\epsilon} = \int_0^T \overline{\lambda(t)}[\frac{dz(t)}{dt} - f(z(t), t, \theta)] dt \tag{18}$$

At optimal $\lambda(t)$, $\frac{dL}{d\epsilon}|_{\epsilon=0} = 0$ for all continuous differentiable $\overline{\lambda(t)}$.

Therefore,

$$\frac{dz(t)}{dt} - f(z(t), t, \theta) = 0, \ \ \forall t \in (0, T) \tag{19}$$

**Derivative w.r.t $z$**  Consider perturbation $\overline{z(t)}$ on $z(t)$, with scale $\epsilon$. With similar analysis:

$$L(\epsilon) = J(z(T) + \epsilon\overline{z(T)}, y) + \int_0^T \lambda(t)[\frac{dz(t) + \epsilon\overline{z(t)}}{dt} - f(z(t) + \epsilon\overline{z(t)}, t, \theta)]dt \qquad (20)$$

Take derivative w.r.t $\epsilon$, it's easy to check conditions for Leibniz integral rule are satisfied, when $f$ and $\overline{z(t)}$ are Lipschitz continuous differentiable functions:

(1) $f(z(t), t, \theta)$ is a Lebesgue-integrable function of $\theta$ for each $z(t) \in \mathbf{R}^d$, since we use a neural network to represent $f$, which is continuous and differentiable almost everywhere.

(2) for almost all $\theta$, $\frac{\partial f(z(t), t, \theta)}{\partial z(t)}$ exists for almost all $x \in \mathbf{R}^d$.

(3) $\frac{\partial f(z(t), t, \theta)}{\partial z(t)}$ is bounded by $g(\theta)$ for all $z(t)$ for almost all $\theta$.

Then we calculate $\frac{dL(\epsilon)}{d}$, note that we can switch integral and derivative:

$$\frac{dL}{d\epsilon}|_{\epsilon=0} = \frac{\partial J}{\partial z(T)}\overline{z(T)} + \frac{d}{d\epsilon}\int_0^T \lambda(t)[\frac{dz(t) + \epsilon\overline{z(t)}}{dt} - f(z(t) + \epsilon\overline{z(t)}, t, \theta)]dt \qquad (21)$$

$$= \frac{\partial J}{\partial z(T)}\overline{z(T)} + \int_0^T \lambda(t)[\frac{d\overline{z(t)}}{dt} - \frac{\partial f(z(t), t, \theta)}{\partial z(t)}\overline{z(t)}]dt \qquad (22)$$

$$= \frac{\partial J}{\partial z(T)}\overline{z(T)} + \int_0^T [\lambda(t)\frac{d\overline{z(t)}}{dt} + \frac{d\lambda(t)}{dt}\overline{z(t)} - \frac{d\lambda(t)}{dt}\overline{z(t)} - \lambda(t)\frac{\partial f(z(t), t, \theta)}{\partial z(t)}\overline{z(t)}]dt \qquad (23)$$

$$= \frac{\partial J}{\partial z(T)}\overline{z(T)} + \lambda(t)\overline{z(t)}|_0^T - \int_0^T \overline{z(t)}[\frac{d\lambda(t)}{dt} + \lambda(t)\frac{\partial f(z(t), t, \theta)}{\partial z(t)}]dt \qquad (24)$$

$$= \frac{\partial J}{\partial z(T)}\overline{z(T)} + \lambda(T)\overline{z(T)} - \lambda(0)\overline{z(0)} - \int_0^T \overline{z(t)}[\frac{d\lambda(t)}{dt} + \lambda(t)\frac{\partial f(z(t), t, \theta)}{\partial z(t)}]dt \qquad (25)$$

$$= (\frac{\partial J}{\partial z(T)} + \lambda(T))\overline{z(T)} - \lambda(0)\overline{z(0)} - \int_0^T \overline{z(t)}[\frac{d\lambda(t)}{dt} + \lambda(t)\frac{\partial f(z(t), t, \theta)}{\partial z(t)}]dt \qquad (26)$$

Since the initial condition $z(0) = x$ is given, perturbation $\overline{z(0)}$ at $t = 0$ is 0, then we have:

$$\frac{dL}{d\epsilon}|_{\epsilon=0} = (\frac{\partial J}{\partial z(T)} + \lambda(T))\overline{z(T)} - \int_0^T \overline{z(t)}[\frac{d\lambda(t)}{dt} + \lambda(t)\frac{\partial f(z(t), t, \theta)}{\partial z(t)}]dt = 0 \qquad (27)$$

for any $\overline{z(t)}$ $s.t.$ $\overline{z(0)} = 0$ and $\overline{z(t)}$ is differentiable.

The solution is:

$$\frac{\partial J}{\partial z(T)} + \lambda(T) = 0 \qquad (28)$$

$$\frac{d\lambda(t)}{dt} + \lambda(t)\frac{\partial f(z(t), t, \theta)}{\partial z(t)} = 0 \ \forall t \in (0, T) \qquad (29)$$

**Derivative w.r.t $\theta$**  From Eq. 16,

$$\frac{dL}{d\theta} = -\int_0^T \lambda(t)\frac{\partial f(z(t), t, \theta)}{\partial \theta}dt \qquad (30)$$

To sum up, first solve the ODE forward-in-time with Eq. 19, then determine the boundary condition by Eq. 28, then solve the ODE backward with Eq. 29, and finally calculate the gradient with Eq. 30. In fact $\lambda$ corresponds to the negative $adjoint$.

**From continuous to discrete case**   To derive corresponding results in discrete cases, we need to replace all integration with finite sum.

In discrete cases, the ODE condition turns into:

$$\frac{z_{i+1} - z_i}{t_{i+1} - t_i} = f(z_i, t_i, \theta) \tag{31}$$

from Eq. 31, we can get:

$$\frac{\partial L}{\partial z_i} = \frac{\partial L}{\partial z_{i+1}} \frac{\partial z_{i+1}}{\partial z_i} = \frac{\partial L}{\partial z_{i+1}} (I + \frac{\partial f(z_i, t_i, \theta)}{\partial z_i} (t_{i+1} - t_i)) \tag{32}$$

Re-arranging terms we have:

$$[(-\frac{\partial L}{\partial z_{i+1}}) - (-\frac{\partial L}{\partial z_i})]/[t_{i+1} - t_i] + (-\frac{\partial L}{\partial z_{i+1}})\frac{\partial f(z_i, t_i, \theta)}{\partial z_i} = 0 \tag{33}$$

which is the discrete version of Eq. 29. Which also corresponds to our analysis in Eq. 10 and 11.

