# OpenReview forum: "Ordinary differential equations on graph networks"
_ICLR.cc/2020/Conference — Reject_

### Official Review · AnonReviewer3 · 2019-10-22
**Official Blind Review #3**

**Rating:** 6

**Review:**

Summary:  This work extends Neural ODE to graph networks and compares the continuous adjoint method with propagation through the ODE solver.

The paper addresses an interesting and important problem and it is well-written in general.
To determine the significance of this work, I have two questions:

Question:
1. What is the major difference between the original Neural ODE and the Graph Neural ODE?
For example, In graph networks, each node’s representation may depend on its neighbor nodes. Will this impact the way you formulate the adjoints or compute the derivative?

2. It seems in Mechanical engineering, various adjoint methods such as Discrete adjoint (e.g. [1])  has been studied.
How does the direct propagation through solver related to the these Discrete adjoint methods?

Some minor comments for experiments:
The authors have 10 runs and take the best one. How about the average? which maybe a better indicator for stability.
How is the runtime comparing normal NN, adjoint, and direct propagation. Runtime has been a major disadvantage for Neural ODE.

Decision:
Overall, the novelty seems somewhat incremental, but I still feel the work is concrete and meaningful. I vote for weak accept.
Looking forward to the code.

[1] A Discrete Adjoint-Based Approach for Optimization Problems on Three-Dimensional Unstructured Meshes. Dimitri J. Mavriplis

**Experience Assessment:**

I have read many papers in this area.

**Review Assessment: Checking Correctness Of Derivations And Theory:**

I assessed the sensibility of the derivations and theory.

**Review Assessment: Checking Correctness Of Experiments:**

I assessed the sensibility of the experiments.

**Review Assessment: Thoroughness In Paper Reading:**

I read the paper at least twice and used my best judgement in assessing the paper.

---

> ### Author Response · Authors · 2019-11-10
> **Response to Reviewer #3**
>
> We sincerely thank the reviewer for your feedback. We address the reviewer’s concerns:
>
> 1.    Difference between neural ODE and graph ODE.
> From the perspective of the model, they are both ODE models $\frac{dz}{dt}=f(z,t)$, except the actual form of $f$ is different. The adjoint and gradient are calculated in the same way.
> Our method is a generic solution and can handle both cases.
>
> 2. Relation to discrete adjoint method
>     To our knowledge, discrete adjoint method is a discretize-then-optimize method, where the continuous function is discretized at pre-defined grids [1]. With pre-defined grids, the continuous problem is actually transformed into a discrete problem with more constraints. This is related to R2’s concern that we are not directly solving the continuous problem.
>     Our method is an optimize-then-discretize approach. The grid points are not predefined, instead they are adaptively computed under error control. This is related to our discussion in thread A.2.
>     For our method, because the grid is adaptively determined, the grid discretization will be very different for different input data; while the grid remains the same for all data in discrete-adjoint methods.
>     Furthermore, because our methods are adaptive, the integration value is robust to different ODE solvers, as long as the error tolerance is small enough. However, with fixed grid, there’s no control on error, hence different solvers could generate very different results.
>     This argument is related to thread A.2, clarifying that our method is truly solving a continuous problem, rather than converting it to a discrete problem.
>
> 3. Experiments
>     The numbers reported in our paper are averages among 10 runs, with standard deviation also reported. We will clarify this in the paper.
>
> 4. Running time
>     For a ResNet18, and its direct modification into an ODE, the discrete model has the shortest running time; the running time of ODE is roughly 4 times longer than the discrete model, when trained both with our method and the adjoint method in [2]. The longer running time comes from the ODE model, where multiple steps need to be determined for numerical integration.
>
> 5. Code and pretrained weights: https://www.dropbox.com/sh/sgaid3efh4eqmjl/AAB-DFXvNq_Pf313UqSLl4VPa?dl=0
>
> 6.    Considering the comments and confusion from all reviewers, we clarify our contributions:
>
> (a)        A general method for ODE solver that works well in practice
>
>         ----Our method is applicable to both CNN-ODE and graph-ODE of different structures, and deals with a fundamental problem: how to estimate the gradient.
>         ----To our knowledge, this is the first paper that makes neural-ODE achieve comparable or even higher accuracy in benchmark classification tasks compared to a state-of-the-art discrete-layer network.
>         -----A general method for free-form functions, and truly designed for continuous models, as discussed in thread A.
>
> (b)    Application to graph networks enables new theoretical aspects, such as over-smoothing phenomena, which is in part discussed in response to R1 (https://openreview.net/forum?id=SJg9z6VFDr&noteId=B1esk0ZHjS )
>
> [1] Nadarajah, Siva, and Antony Jameson. "Studies of the continuous and discrete adjoint approaches to viscous automatic aerodynamic shape optimization."2001.
> [2] Chen, Tian Qi, et al. "Neural ordinary differential equations." Advances in neural information processing systems. 2018.

---

### Official Review · AnonReviewer1 · 2019-10-23
**Official Blind Review #1**

**Rating:** 3

**Review:**

Summary

The authors discussed that most graph NNs to date considered discrete layers and hence are difficult to model diffusion processes on graphs. This paper proposed the Neural ODE on a graph, termed with a graph ODE, to tackle this problem. The authors gave a sufficient condition under which the adjoint method gets instable and pointed out potential issues of training Neural ODEs using it. To overcome the instability issue, the author proposed to backpropagate errors directly at discretized points. Since the naive implementation of the direct method is memory-consuming, the authors used invertible blocks as building blocks of graph ODEs, which do not store the intermediate activations for backward propagation. Finally, the authors conducted empirical studies to see the effectiveness of the proposed method.


Decision

I recommend rejecting the paper weakly because I think the extension of Neural ODEs to graphs is straightforward and that the empirical study is not strong enough to compensate for the weakness of the novelty.
Theoretical justification of numerical instability of Neural ODEs (Proposition 1) and its empirical verification (Section 5.4) give new insights for understanding Neural ODEs. However, if I understand correctly, the formulation of graph ODEs do not use the internal structures of graph NNs, even the fact that the underlying neural network is a graph NN. Therefore, I think the extension from Neural ODEs to graphs is a bit too straightforward. The authors proposed a method which improves the stability and memory-efficiency of training. We can apply this method to general Neural ODEs, too. In addition, we can attribute the idea of using invertible blocks to existing works (Gomez et al., 2017). Finally, regarding the empirical performance of graph ODEs, the performance improvement from existing GNNs is within the standard deviations. Therefore, I think the empirical result is not sufficiently strong to justify the novelty of applying Neural ODEs to graphs. Taking these things into account, although the authors gave a new result on Neural ODEs, I think the contribution is limited from the viewpoint of the study of graph NNs.


Suggestion

- As I wrote in the Decision section, the theoretical results are not restricted to graph ODEs but valid for general neural ODEs. Therefore, I think the authors do not have to restrict the application areas to graph ODEs. The possible direction of the paper is to further analysis of training neural ODEs (e.g., instability). On the other hand, if the authors are interested in the extension of neural ODEs to graphs, I expect a more detailed relationship between the neural ODEs framework and underlying GNNs. For example, I am curious how the topological information of graphs affects the graph ODEs via spectral-type graph convolution operations and what is the relationship to the oversmoothing phenomena (Li et al., 2018).
- Since Theorem 1 is applicable not only graph ODEs but also Neural ODEs, it implies that ordinal Neural ODEs are also vulnerable to the instability. The experiments in Dupont (2019), which this paper referenced, pointed out the instability of Neural ODEs. I am wondering the proposed method can enhance the training of neural ODEs, too.

[Li et al., 2018] Li, Qimai, Zhichao Han, and Xiao-Ming Wu. "Deeper insights into graph convolutional networks for semi-supervised learning." Thirty-Second AAAI Conference on Artificial Intelligence. 2018.

**Experience Assessment:**

I have published one or two papers in this area.

**Review Assessment: Checking Correctness Of Derivations And Theory:**

I assessed the sensibility of the derivations and theory.

**Review Assessment: Checking Correctness Of Experiments:**

I assessed the sensibility of the experiments.

**Review Assessment: Thoroughness In Paper Reading:**

I read the paper at least twice and used my best judgement in assessing the paper.

---

> ### Author Response · Authors · 2019-11-10
> **Response to Reviewer # 1**
>
> We sincerely thank the reviewer for your efforts and comments. We will address the reviewer’s concerns and state our contributions:
>
> 1. Contribution to graph neural networks, connection with over-smoothing phenomena
>
> -----(a) intuition
> We sincerely thank the reviewer for pointing out that ODE and the over-smoothing issue of GCN could be related. In fact, we did not start this project with neural ODE; instead we started with an intuition that GNN could be related to a continuous “heat transfer” model, then we tried to build a good model for the continuous process, so we started looking at ODEs but found current method can not train an ODE well.
>
> In the heat-transfer model, the feature of each node is analogous to temperature, and the messaging passing process is analogous to heat transfer between nodes. Time of heat transfer process is analogous to depths of discrete-layer GCN models. If time is too short, there will be insufficient heat transfer, equivalent to a node not having enough information from its neighbors to update; if time is too long, all nodes end up in the same temperature, corresponding to the over-smoothing issue with GNN.
>
> ----(b) Mathematical analysis
> We sincerely thank the reviewer for pointing out previous work on over-smoothing, which leads us from intuition to some mathematical analysis. We show our analysis below:
>
> It’s demonstrated that graph convolution is a special case of Laplacian smoothing in [1], which can be written as
> $Y = (I - \gamma \tilde{D}^{-1} \tilde{L})X$
> where $X$ and $Y$ are the input and output of a graph-conv layer respectively, $\tilde{A}=A+I$ where $A$ is the adjacency matrix, and $\tilde{D}$ is the corresponding degree matrix of $\tilde{A}$, and $\gamma$ is a positive scaling constant.
> If we replace normalized Laplacian with symmetricly normalized Laplacian, the continuous process becomes:
> $Y = (I - \gamma \tilde{D}^{-1/2} \tilde{L} \tilde{D}^{-1/2} )X$
> When it’s modified from a discrete model to a continuous model, the continuous smoothing process is:
> $\frac{dX}{dt} = - \gamma \tilde{D}^{-1/2} \tilde{L} \tilde{D}^{-1/2} X$
> Since all eigenvalues of the symmetrically normalized Laplacian are real and non-negative, then all eigenvalues of the above ODE are real and non-positive.
> Suppose all eigenvalues of the normalized Laplacian are non-zero. In this case, the ODE has only negative eigenvalues, hence the ODE above is $asymptotically \ stable$ [2]. Hence as time $t$ grows sufficiently large, all trajectories are close enough.
> In experiments, this suggests if integration time $T$ is large enough, all nodes (from different classes) will have very similar features, thus the classification accuracy will drop.
>
> ----(c) Experiments
> We validate this with experiments. Node classification accuracy varying with integration time is measured on the Cora dataset. Results are shown below:
> —————————————————————————————————
> Integration time    0.5      1.0       1.5       2.0       5.0      10.0     20.0    100.0
> —————————————————————————————————
> Accuracy                  80.5   81.6     80.1    80.1     79.3     77.6     68.9     35.1
> —————————————————————————————————
>
> When integration time is small (0.5), nodes do not aggregate neighbor information. When integration time is too long, node attributes tend to the same value because of asymptotic behaviour of ODE, as mentioned above. Both cases generate inferior accuracy.
>
> 2.  Generalization to general ODEs.
>      We show that our method can be applied to general ODEs, with free-form functions, can deal with continuous models, and are robust to different ODE solvers. Details are in appendix C of original version (Sec 5.3 in revision), and we post it in thread A (https://openreview.net/forum?id=SJg9z6VFDr&noteId=B1eeWWA7sr ).
>
> 3.  Our method supports free-form functions and does not depend on invertible block
>     Details are in thread B (https://openreview.net/forum?id=SJg9z6VFDr&noteId=SkgzfxCmoH ).  Our method is memory efficient for free-form functions, with the special case of invertible block we achieve even lower memory. Hence the contribution cannot be simply attributed to invertible block.
>
> 4.  Better performance for both graph network and CNN
>      We validate our method both in CNN-ODE (thread A.3) and GNN-ODE (thread C https://openreview.net/forum?id=SJg9z6VFDr&noteId=HklsTJ07jB ). The CNN-ODE is a *free-form* function; for GNN-ODE, we tested both free-form and invertible block (restricted form). Our method significantly outperforms (under paired t-test) discrete models for graph networks; it reduces error rate from 19% to 5% on CIFAR10 compared with results using the adjoint method [3] and Augmented-NODE[4].
>
>
> [1] Li, Qimai, et al. "Deeper insights into graph convolutional networks for semi-supervised learning"2018.
> [2] Lyapunov, A. M. The General Problem of the Stability of Motion
> [3] Chen, Tian Qi, et al. "Neural ordinary differential equations." 2018
> [4] Dupont, et al. "Augmented neural odes." (2019).

---

> > ### Comment · AnonReviewer1 · 2019-11-13
> > **Oversmoothing of graph ODEs.**
> >
> > Thank you for discussing the relationship between graph ODEs and the oversmoothing phenomena and sharing the additional experiment results.
> >
> > It is interesting that the performance consistently drops as the integration time increases. Although the model authors used do not have non-linear activation functions, as opposed to usual GCNs, it seems that this is supporting evidence that oversmoothing happens to graph ODEs, too. Considering that the resulting model is a diffusion process on graphs, I guess we can interpret that the process converges to an equilibrium.
> >
> > Taking this experiments into account, it would be great if there is a suggestion on how to overcome the oversmoothing phenomena through the lens of ODEs.

---

> > > ### Author Response · Authors · 2019-11-13
> > > **Potential direction to overcome oversmoothing for graph ODE**
> > >
> > > Thanks for your discussion, we agree that the oversmoothing phenomena on graph ODE can be interpreted as the ODE systems converge to equilibrium. From this point, here are several potential solutions:
> > >
> > > 1.    Limit the integration time to a small value, since the convergence phenomena occurs when time $t$ goes sufficiently large.
> > >
> > > 2.   Limit the amplitude of the eigenvalue of $f$.
> > > Ignoring the constant due to initial value, the solution to a linear ODE system $\frac{dX}{dt}=AX$ is $X(t) = e^{At}X(0)$
> > > Writing in the form of eigenvalue $\lambda_i$ and eigenvectors $u_i$ of matrix $A$, the solution is:
> > > $X(t) = c_1 e^{\lambda_1}u_1 + c_2 e^{\lambda_2}u_2 + ...$.
> > >
> > > For GCN-ODE specifically, all eigenvalues are non-positive. The larger $\vert \lambda_i \vert$, the faster $X(t)$ decays, and the system reaches equilibrium. Reducing $\vert \lambda \vert$ (or limit the Lipschitz constant of $A$) is a possible solution, which can potentially be achieved by:
> > >
> > > (1)  Clip weights within a fixed range to control the Lipschitz constant as in Wasserstein GAN [1]
> > > (2)  Estimate the spectral norm of $A$ and limit it in the loss function. In practice, the bound on spectral norm can be estimated with power iterations. [2]
> > >
> > > [1] Arjovsky, Martin, Soumith Chintala, and Léon Bottou. "Wasserstein gan." arXiv preprint arXiv:1701.07875 (2017).
> > > [2] Miyato, Takeru, et al. "Spectral normalization for generative adversarial networks." arXiv preprint arXiv:1802.05957 (2018).

---

### Official Review · AnonReviewer2 · 2019-10-24
**Official Blind Review #2**

**Rating:** 1

**Review:**

Summary: This paper proposed a deep model called graph-ODE (GODE), which extends the continuous deep model, neural ODE [1], to graph structured data. Two methods of computing the gradient are proposed, one is the adjoint method proposed in [1] which is memory-efficient but not perfectly accurate, and the other method is to discretize the ODE and compute an accurate gradient, which is also memory-efficient but relies on an invertible architecture.

In general, this paper contains many interesting thoughts, but to me, it seems these ideas are not new and have been proposed before[1,2,3,4]. I could not find a strong contribution of this paper.

- - - about the ODE stability issue

Among invertible deep models, the advantage of ODE-based continuous models [1] is: there is *no restriction* on the form of f. The drawback is the computed gradient has an error, depending on discretization step size and stability. This paper pointed out the stability problem (which is also discussed in [2,3]), but do not provide a solution in the domain of continuous deep models.

Instead, the solution they provided is to use a discretized version and compute the gradient accurately. Then it becomes a standard invertible DL model with discrete layers, where the invertible building blocks have a specific *restricted form*. The 'adjoint method with discrete-time' in Eq (10) is the same as the chain rule, which has also be pointed out in [4]. To this point, I think GODE is in the class of discrete invertible DL models trained by gradient descent. I think it less related to continuous models, except the step size can be adaptive in the forward pass.

- - - about the invertible building block

The proposed invertible building block replaces 'sum' in [5] by a function \psi. This is not novel enough to serve as a contribution.

- - - comparison to graph neural network

I think it is interesting to apply ODE techniques or other invertible architectures to graph-structured data, for which I didn't see similar works before and could be a contribution of this paper. However, for the experimental results shown in table 3 and 4, the improvement is really small. A stronger result is needed to demonstrate the advantages.

[1] Chen, Tian Qi, et al. "Neural ordinary differential equations." Advances in neural information processing systems. 2018.
[2] Chang, Bo, et al. "Reversible architectures for arbitrarily deep residual neural networks." Thirty-Second AAAI Conference on Artificial Intelligence. 2018.
[3] Behrmann, Jens, et al. "Invertible Residual Networks." International Conference on Machine Learning. 2019.
[4] Li, Qianxiao, et al. "Maximum principle based algorithms for deep learning." The Journal of Machine Learning Research 18.1 (2017): 5998-6026.
[5] Gomez, Aidan N., et al. "The reversible residual network: Backpropagation without storing activations." Advances in neural information processing systems. 2017.



-------------after reading the response

I'd like to thank the authors for their explanations.
However, the authors' explanation of the benefit of transforming the invertible building block into an ODE model is still not convincing to me.

The authors explained that ODE solutions can represent highly nonlinear functions. However, discrete NN also represents highly nonlinear functions (e.g. with Relu activation, they are *piecewise* linear, but they are highly nonlinear)! From my point of view, their difference is that ODE model is more smooth. However, the benefit of using a smoother model is still unclear. For the example that the authors provided, $\sin x$, why being able to represent that kind of function is an advantage for the graph classification problem? Why is this a good model bias? I think the authors' responses are still not convincing enough, so I choose to retain the score.

**Experience Assessment:**

I have published one or two papers in this area.

**Review Assessment: Checking Correctness Of Derivations And Theory:**

I assessed the sensibility of the derivations and theory.

**Review Assessment: Checking Correctness Of Experiments:**

I assessed the sensibility of the experiments.

**Review Assessment: Thoroughness In Paper Reading:**

I read the paper at least twice and used my best judgement in assessing the paper.

---

> ### Author Response · Authors · 2019-11-09
> **Response to Reviewer #2**
>
> We sincerely thank you for review and totally agree with your comments on ODE solver for free-form continuous functions. However, we want to point out that our method is a free-form solver for continuous models, and extensive experiments are conducted in appendix C (moved to Sec 5.3 in revision). The reviewer might overlooked this part because we did not put enough emphasis in the main paper. We will address your concerns in the following.
>
> 1.         about the ODE stability issue
> We put a detailed description on this issue in thread A https://openreview.net/forum?id=SJg9z6VFDr&noteId=B1eeWWA7sr
> We will briefly address your concern here.
>
> (a) Our method can handle free-form functions, which is theoretically discussed in thread A.1. For experiments in thread A.3, we directly modified a ResNet18 into an ODE, where $f$ is a stack of conv-bn-relu-conv-bn layers WITHOUT the invertible structure.
>
> In fact, invertible block is not the key to our ODE solver; it’s only the key to low memory consumption. This is discussed in thread B, section 1 to 3 https://openreview.net/forum?id=SJg9z6VFDr&noteId=SkgzfxCmoH
> Memory consumption for naive direct backprop is $O(N_f\times N_t\times K)$, our method for free-form can be reduced to $O(N_f + N_t)$, for invertible blocks is $O(N_f)$
>
> We apologize for the confusion; we put too little description of the ability to handle free-form, and wrote too much about invertible block with the thinking that most readers might not be familiar. We will revise this in the paper.
>
> (b) Our method is dealing with a continuous model.
> This is theoretically discussed in thread A.2 and supported with experiments in A.3.(d).
>
> We agree with reviewers that uncareful discretization will convert a continuous model into a discrete model. However, we clarify that this is not the case with our method.
>
> A continuous function is robust to different ODE solvers, while discrete models are sensitive to solvers, because different ODE solvers are equivalent to different depths.
>
> A test for continuous model is: at inference time, switch to different ODE solvers of different orders; a continuous model outputs very robust results.
>
> Our method passed this test, as shown in thread A.3.(d). You can check this with our code and pre-trained weights https://www.dropbox.com/sh/sgaid3efh4eqmjl/AAB-DFXvNq_Pf313UqSLl4VPa?dl=0
>
>
> 2.   about the invertible building block
>     We admit that we did not extensively experiment with different forms of $\psi$; we will revise our paper and not claim this as our main contribution. However, we want to clarify that we define a space to search for invertible blocks, which is important for the field of “normalizing flow”. There are many works on normalizing flow which uses different $\psi$. For example, $\psi(\alpha,\beta) = \alpha \cdot sigmoid(\beta)$ in [2], $\psi(\alpha,\beta) = \alpha + \beta$ in [3], $\psi(\alpha,\beta) = \alpha \cdot exp(\beta)$ in [4], $\psi(\alpha,\beta) = \alpha \cdot \beta$ in [5]. To our knowledge, we are the first to define the searching space of $\psi$.
>
>
> 3.    empirical performance
>      We validate our method both in CNN-ODE (thread A.3) and GNN-ODE (thread C). The CNN-ODE is a *free-form* function; for GNN-ODE, we tested both free-form and invertible block (restricted form).
>
>      For experiments on CIFAR with CNN-ODE, our method reduces error rate from 19% to 5% compared with training ODE using the solver in [1]. Furthermore, our model is robust to different ODE solvers during inference. Finally, our model has the same number of parameters as a ResNet18, but outperforms standard ResNet50 and ResNet101.
>
>    For experiments with GNN, we re-run experiments and performed paired t-tests. Results are summarized in Thread C ( https://openreview.net/forum?id=SJg9z6VFDr&noteId=HklsTJ07jB  ). For most cases, our model outperforms the discrete baseline at a 1% significance level.
>
>
> [1] Chen, Tian Qi, et al. "Neural ordinary differential equations." Advances in neural information processing systems. 2018.
> [2] Huang, Chin-Wei, et al. "Neural autoregressive flows." arXiv preprint arXiv:1804.00779 (2018).
> [3] Dinh, Laurent, David Krueger, and Yoshua Bengio. "Nice: Non-linear independent components estimation." arXiv preprint arXiv:1410.8516 (2014).
> [4] Dinh, Laurent, Jascha Sohl-Dickstein, and Samy Bengio. "Density estimation using real nvp." arXiv preprint arXiv:1605.08803 (2016).
> [5] Liao, Huadong, Jiawei He, and Kunxian Shu. "Generative Model with Dynamic Linear Flow." arXiv preprint arXiv:1905.03239 (2019).

---

> > ### Comment · AnonReviewer2 · 2019-11-12
> > **For the invertible building block version, what is the benefit of modeling it as a solution of ODE?**
> >
> > Now my understanding is the proposed method has two versions:
> > (1) the free-form continuous version.
> > (2) restricted form, and compute the gradient exactly.
> >
> > For version (1), I do not have many questions. It is the same as the Neural ODE approach, but applied to Graph data. I think if the experimental results are very good, then this new application can be counted as a contribution.
> >
> > Version (2) constitutes the major contents in this paper, but I couldn't see how we benefit from viewing it as an ODE model.
> >
> > To be more clear, we need to notice that we are solving *a machine learning problem*  (e.g. node classification or graph classification) *not solving an ODE*. Why do we need to use a small step size in this case? Why do we need to make the output as an approximate solution of ODE? Why is it a good inductive bias? I will think using a larger step size and not approximating the solution of ODE (just like what other discrete invertible models do) may be even better.
> >
> > I think the advantages/disadvantages of version (2) are the same as other discrete models: memory-efficient but restricted form. I personally think version (2) is more like a discrete model because I don't see the additional benefit of viewing it as an ODE in this machine learning problem.
> >
> > Could the authors explain a bit about:
> >
> > For version (2), what's the benefit of modeling it as a solution of ODE? Wouldn't it be better to choose a larger step size just like in other discrete models?
> >
> > For version (1), the benefit is clear to me: free-form.

---

> > > ### Author Response · Authors · 2019-11-13
> > > **Benefits of modeling invertible blocks as ODE (Part 1/2)**
> > >
> > > Thank you for the comments, we explain the benefits to model invertible blocks as ODEs.
> > >
> > > 1.   In terms of practical performance, ODE version of invertible blocks has the potential to outperform discrete-layer version if it's well trained, which is shown in table of thread C. (We also observe NODE18 outperforms ResNet101 in image classification in thread A)
> > >
> > > This could be because when modifying an invertible block into an ODE, the model has the potential to capture some highly nonlinear functions. For example, it takes many layers of discrete model with ReLU activations to capture the function $y=sint$, because ReLU activations make the network behave piecewise linear; but it can be captured by a 1-layer ODE,
> > > $$
> > > \begin{bmatrix}
> > > \frac{da}{dt} \\ \frac{db}{dt}
> > > \end{bmatrix}
> > > = \begin{bmatrix}   0& 1\\ -1 & 0 \end{bmatrix}  \begin{bmatrix} a \\ b  \end{bmatrix}
> > > $$
> > > Where $a = y, b = \frac{dy}{dt}$.
> > >
> > > 2.    Other benefits include:
> > > (1) Explicit error estimation with most ODE solvers, so the output error estimation of ODE can be used to determine decision confidence.
> > > (2) Speed-accuracy trade-off by controlling error tolerance.
> > > (3) Robustness to input perturbation and adversarial attack, which is empirically and theoretically analyzed in [1]
> > >
> > > [1] Yan, Hanshu, et al. "On Robustness of Neural Ordinary Differential Equations." arXiv preprint arXiv:1910.05513 (2019).

---

> > > ### Author Response · Authors · 2019-11-14
> > > **Clarification on contribution to accurate gradient estimation in version (1)  (Part 2/2):**
> > >
> > > Thank you for pointing out the importance of free-form for version (1). We want to clarify our contributions in version (1):
> > > It’s not only a new application to graph data, but also a generic method to accurately estimate the gradient for neural-ODEs. This is a fundamental problem to deep learning models, which is not extensively studied in the original Neural ODE paper.
> > >
> > > Please see end of intro in the revised paper for an updated list of our contribution.

---

### Author Response · Authors · 2019-11-09
**C:   Empirical performance of graph-ode**

We thank all reviewers for their comments on the empirical performance. We re-run the experiments on graph networks, with both invertible blocks and free-form functions (as suggested by R2). We performed a paired t-test. We use * (**) to mark GODE models that outperform corresponding discrete-layer baselines at a 5% (1%) significance level.

Accuracy on a 10-fold cross validation is reported below. We will also update results in the main paper. Discrete layer models are marked with “DISC”, GODE with invertible blocks are marked with “restrict”, and with free-form functions are marked with “free”. For various baselines, most of the GODE models significantly outperform the baseline.
                         Accuracy(%) in graph-classification tasks
+------------------------------+----------------+------------------+-----------------+---------------+
| Model                           | MUTAG       | PROTEINS    | IMDB          | REDDIT      |
+---------+-------------------+-----------------+------------------+----------------+----------------+
| GCN  | DISC                 | 73.3+-5.2    | 72.4+-3.1       | 74.2+-3.6   | 85.9+-1.8   |
|           +--------+----------+-----------------+------------------+----------------+---------------+
|           | ODE | restrict |74.7+-4.3** | 74.7+-4.3** | 75.3+-5.3*  | 89.2+-3.2** |
|           |           +----------+-----------------+------------------+----------------+----------------+
|           |           | free     | 75.1+-5.3** | 76.6+-3.9** | 73.9+-4.6   | 88.5+-3.0** |
+---------+--------+----------+-----------------+------------------+----------------+-------------+
| Cheb | DISC                 | 84.0+-6.4    | 70.6+-3.9       | 71.9+-3.8   | 91.0+-1.5   |
|            +--------+----------+----------------+-------------------+----------------+-------------+
|            | ODE | restrict | 85.0+-8.3   | 72.7+-3.7** | 72.0+-5.0*  | 91.2+-1.5   |
|            |          +----------+-----------------+-----------------+-----------------+-------------+
|            |          | free     | 86.1+-6.3*  | 72.5+-4.7** | 73.6+-4.0** | 92.4+-1.6** |
+---------+--------+----------+-----------------+-----------------+-----------------+---------------+
| GIN    | DISC                | 85.0+-6.4    | 73.0+-3.1      | 73.3+-5.1     | 89.2+-2.5   |
|            +--------+----------+-----------------+-----------------+-----------------+-------------+
|            | ODE | restrict | 87.7+-5.2*  | 74.5+-3.5*  | 75.4+-3.9** | 90.5+-1.5** |
|            |           +----------+-----------------+----------------+------------------+-------------+
|            |           | free     | 87.8+-5.1** | 73.1+-3.7   | 74.8+-4.1** | 91.0+-2.4** |
+----------+-------+----------+------------------+----------------+-----------------+-------------+

---

### Author Response · Authors · 2019-11-09
**B:  Details of algorithm for free-form ODE with accurate gradient estimation**

In the original submission in Appendix B (moved to Sec 3.3 in revision) we detailed the algorithm for using invertible blocks, the most memory efficient form of our solver. In the revision we will include the following algorithm for free-form models.

+ —————————————————————————————————————————————+
Define model $\frac{\mathrm{d} z(t)}{ \mathrm{d} t} = f(z(t), t)$, where $f$ is a free-form function. Denote integration time as $T$.
Forward ($f, T, z_0, tolerance$):
         $t=0, z = z_0$
         $state_0 = f.state\_dict()$, cache.save($state_0$)
         Select initial step size $h=h_0$ (adaptively with adaptive step-size solver).
         $time\_points = empty\_list()$
         While $t<T$:
                $state = f.state\_dict()$,   $accept\_step = False$
                While Not  $accept\_step$:
                            $f.load\_state\_dict(state)$
                            with $grad\_disabled$:
                                     $z\_new, error\_estimate = step(f, z, t, h)$
                            If $error\_estimate < tolerance$:
                                    $accept\_step = True$
                                    $z = z\_new,   t = t + h,   time\_points.append(t)$
                            else:
                                     reduce stepsize $h$ according to $error\_estimate$
                                     delete $z\_new, error\_estimate$ and related local computation graph
        cache.save($time\_points$)
        return $z$, cache
+————--————————————————————————————————————————-+
 Backward ($f, T, z_0, tolerance, cache$):
         $\{t_0, t_1, t_2, ...t_{N-1}, t_N\} = cache.time\_points$
         For $t_i$ in $\{t_0, t_1, t_2, ...t_{N-1}, t_N\}:$
                 $z(t_{i+1}) = step(f, z(t_i), t_i, h=t_{i+1} - t_i)$
         For $t_i$ in $\{ t_N, t_{N-1}, ..., t_1, t_0 \}:$
                  Determine $a(t_i)=\frac{\partial L}{\partial z(t_i)}$ and $\frac{\partial z(t_i)}{\partial \theta}$
         $\frac{\mathrm{d} L}{\mathrm{d} \theta} = \sum_{i=1}^{N} a_i \frac{\partial z(t_i)}{ \partial \theta}$
        return $\frac{\mathrm{d} L}{\mathrm{d} \theta}$
+------------—————————————————————————————————————————-+


Memory consumption analysis:
1. Suppose $f$ has $N_f$ layers, the number of forward evaluation step is $N_t$ on average, and the evaluations to adaptively search for an optimal stepsize is $K$. A naive solver will take $O(N_f \times N_t \times K)$, while our method consumes $O(N_f \times N_t)$ because all middle activations are deleted during forward pass, and we don’t need to search for optimal stepsize in backward pass.

2. In fact, if we perform step-wise checkpoint method, where we only store $z(t_i)$ for all $t_i$, and compute the gradient $\frac{\partial z(t_{i+1})}{\partial z(t_i)}$ for one $t_i$ at a time, then the memory consumption can be reduced to $O(N_f + N_t)$.

3. Since the solver can handle free-form functions, it can also handle restricted form invertible block. In this case, we don’t need to store $z(t_i)$, the memory consumption can reduce to $O(N_f)$.  In fact we use invertible blocks only for memory consideration, it does not imply invertible blocks are necessary.

---

### Author Response · Authors · 2019-11-09
**A:  Accurate gradient estimation in ODE solver for free-form functions, with application to general NODEs**

We sincerely thank all reviewers for their insightful comments, and agree on the importance of free-form ODE solver for continuous models, and the application to general ODEs, as mentioned by all reviewers. We want to point out this is exactly covered in appendix C in the original submission (Sec 5.3 in revision), but reviewers might overlook this part because we did not put sufficient emphasis in the main paper. Here we clarify our contributions. We will also revise our paper for better understanding.

1. An ODE solver for free-form functions. Since it can handle free-form functions, it can handle restricted forms. In fact we use invertible blocks only for memory consideration, it does not imply invertible blocks are necessary.
The details of our free-form ODE solver is in thread B(https://openreview.net/forum?id=SJg9z6VFDr&noteId=SkgzfxCmoH ), here we briefly summarize it:
-- During forward pass, the solver performs a numerical integration, with the stepsize adaptively varying with error estimation.
-- During forward pass, the solver outputs integrated value, and the *evaluation time points* {$t_0, t_1, … t_N$}. All middle activations are deleted to save memory.
-- During backward pass, the solver re-builds the computation graph, by *directly* evaluating at saved time points, without adaptive searching.
-- During backward pass, the solver is performing a numerical reverse-time integration $\frac{\mathrm{d} L}{\mathrm{d} \theta} = - \int_T^0 a(t)^T \frac{\partial f(z(t), t, \theta)}{\partial \theta} \mathrm{d}t$.
-- Here we emphasize $z(t_i)$ is accurate, because the backward pass uses the same ${t_i}$ as in the forward pass. Therefore, the adjoint $a(t_i)$ and gradient can be accurately determined.
-- Our method does not require $f$ to be invertible, it can be free-form.

2. A solver truly for continuous models
    We strongly agree with reviewers that uncareful discretization will turn a continuous model into a discrete model. However, we clarify our model is a truly continuous model:
----(a) Numerical discretization in ODE solvers does NOT turn a continuous model into a discrete model. To our knowledge, all existing numerical ODE solvers have a discretization step. They are widely used in physical and mathematical problems such as fluid dynamics, but researchers are still dealing with continuous problems.

----(b) Continuous models are robust to solvers.
A naive fixed stepsize discretization will turn a continuous model into a discrete model. For example, a 1-step discretization is equivalent to a residual block; a 2-step discretization is equivalent to reusing every layer twice. During inference, any change to the discretization is equivalent to changing model depth; this will cause severe error for discrete models.

However, for a continuous model, during inference, different solvers generate very close results as long as the error tolerance is small. Hence, continuous models remain robust to solvers of different orders.  (we call this "robustness test")

We demonstrate our method passes robustness test, shown below in A.3

---

> ### Author Response · Authors · 2019-11-09
> **Empirical performance of free-form CNN-ODE on image classification**
>
>
> +---------------+-----------------------------------------------------------------+-------------------------------------------------+
> |                    |                             ours                                                | Literature                                             |
> |                    +------------------------------------+----------------------------+-------------+--------+-----------+-----------+
> |                    | Adaptive Solvers                | Fixed step solvers   | adjoint   |Res18  |Res50   | Res101 |
> |                    +--------------+---------+----------+--------+--------+--------+                |             |              |              |
> |                    |HeunEuler| RK23  | RK45 | Euler | RK2   | RK4   |                |             |             |               |
> +----------------+-------------+---------+---------+---------+--------+---------+------------+----------+----------+-----------+
> | CIFAR10    | 4.85         | 4.92    | 5.29   | 5.52   | 5.27    | 5.24   | 19.2       | 6.98     | 6.38     | 6.25      |
> +---------------+-------------+---------+---------+---------+---------+---------+------------+--------- -+----------+-----------+
> | CIFAR100  | 22.66      | 24.13 | 23.56 | 24.44  | 24.44 | 24.43 | 37.6        | 27.08   | 25.73   | 24.84    |
> +---------------+-------------+---------+---------+---------+---------+---------+------------+----------+-----------+-----------+
>                                                           Error Rate % on CIFAR10 and CIFAR100 test set
> 3.  Applications to a convolutional neural-ODE (code and pretrained weights: https://www.dropbox.com/sh/sgaid3efh4eqmjl/AAB-DFXvNq_Pf313UqSLl4VPa?dl=0 )
>      We explain the contents in the appendix of the original submission, with results shown in the table above. We will move these results to the main text in the revision.
>
> ----(a)We directly modify a ResNet18 into a corresponding ODE. In a residual block, the forward pass is $y = x + f(x)$; when directly modified into ODE, the forward pass is $y(T) = y_0 + \int_0^T f(y) dt$.
>
> ----(b)Since it’s a direct modification from ResNet, $f$ is $conv-bn-relu-conv-bn$, the same as the residual branch in a ResNet, and it is a *free-form* function.
>
> ----(c)Accurate gradient estimation is the key to high performance
> We trained a single model with the HeunEuler solver (adaptive), and tested with several different solvers. Our model modified achieved 4.85% and 22.66% error rate on CIFAR10 and CIFAR100 respectively, it has the same number of parameters as a ResNet18, but outperforms a standard ResNet101.
>
> Our model outperforms ODE trained with the adjoint method (reported in literature) by a large margin (75% reduction in error rate on CIFAR10), validating the importance of accurate gradient estimation.
>
> ----(d) A continuous model is robust to different solvers.
> For the same model, we trained it with the HeunEuler solver, but tested with different solvers (of different orders) during inference. We observed only 1% increase in error rate. This implies that our model is robust to different equivalent depths caused by different order of solvers, as discussed in 2(b).

---

### Author Response · Authors · 2019-11-13
**Paper revision**

We edit our abstract and intro, and re-order contents for better clarity on the generalizability of our method to general free-form NODE, and contribution to a fundamental problem: accurate gradient estimation.

1.    Re-order of contents in the original submission:
Section in revision    Section in original submission                        Contents
3.1                                    3.2                                                    From discrete to continuous models
3.2                                    4.1                                                    Analysis of adjoint method
3.3 (Algo 1)               4.2 (Algo  2 in Appendix B)                Our method for accurate gradient estimation
4.1                                   3.1                                                     Message passing in GNN
5.3                                 Appendix C                                       Application to general NODE
Appendix B                   4.3                                                     Invertible blocks

2.    Modification of contents:
 (1)  We added paired t-test results to table 3 and 4
 (2)  We addressed the relation to the over-smooth issue in Sec4.2 and Table 5

---

### Decision · Program_Chairs · 2019-12-19

**Decision:**

Reject

**Comment:**

This paper introduces a few ideas to potentially improve the performance of neural ODEs on graph networks.  However, the reviewers disagreed about the motivations for the proposed modifications.  Specifically, it's not clear that neural ODEs provide a more advantageous parameterization in this setting than standard discrete networks.

It's also not clear at all why the authors are discussion graph neural networks in particular, as all of their proposed changes would apply to all types of network.

Another major problem I had with this paper was the assertion that the running the original system backwards leads to large numerical error.  This is a plausible claim, but it was never verified.  It's extremely easy to check (e.g. by comparing the reconstructed initial state at t0 with the true original state at t0, or by comparing gradients computed by different methods).  It's also not clear if the authors enforced the constraints on their dynamics function needed to ensure that a unique solution exists in the first place.